# A Synbiotic Combining Chitin–Glucan and *Lactobacillus acidophilus* NCFM Induces a Colonic Molecular Signature Soothing Intestinal Pain and Inflammation in an Animal Model of IBS

**DOI:** 10.3390/ijms251910732

**Published:** 2024-10-05

**Authors:** Lena Capirchio, Christel Rousseaux, Caroline Dubuquoy, Arthur C. Ouwehand, Véronique Maquet, Salvatore Modica, Edouard Louis, Pierre Desreumaux, Jan Tack

**Affiliations:** 1Hepato-Gastroenterology Department, Centre Hospitalier Wallonie-Picarde, 7500 Tournai, Belgium; lena.capirchio@chwapi.be; 2Intestinal Biotech Development, 59045 Lille, France; crousseaux@ibd-biotech.com (C.R.); carodubu@yahoo.fr (C.D.); 3Health Sciences, IFF Health, 02460 Kantvik, Finland; arthur.ouwehand@iff.com; 4KitoZyme SA, Parc Industriel des Hauts Sarts Zone 2, Rue de Milmort 680, 4040 Herstal, Belgium; v.maquet@kitozyme.com (V.M.); s.modica@biokuris.com (S.M.); 5Department of Gastroenterology, Centre Hospitalier Universitaire de Liège, 4000 Liège, Belgium; edouard.louis@ulg.ac.be; 6U1286—INFINITE—Institute for Translational Research in Inflammation, University Lille, Inserm, CHU Lille, 59000 Lille, France; 7Hepato-Gastroenterology Department, Lille University Hospital, 59000 Lille, France; 8Translational Research Center for Gastrointestinal Disorders (TARGID), KU Leuven, 3000 Leuven, Belgium; jan.tack@kuleuven.be; 9Department of Molecular and Clinical Medicine, Institute of Medicine, Sahlgrenska Academy, University of Gothenburg, 40530 Gothenburg, Sweden; 10Rome Foundation, Raleigh, NC 27614, USA

**Keywords:** chitin-glucan, *Lactobacillus acidophilus* NCFM, probiotic, prebiotic, synbiotic, irritable bowel syndrome, abdominal pain, inflammation, analgesia

## Abstract

Chitin–glucan (CG) is a new generation of prebiotic. *Lactobacillus acidophilus* NCFM^®^ (NCFM) is a probiotic with the ability to decrease abdominal pain. We evaluate the functional and molecular gastrointestinal responses to a synbiotic administration combining CG and NCFM in a rat model of long-lasting colon hypersensitivity. The intracolonic pressure was assessed during the 9-week experiment in animals receiving CG in association or not with NCFM and compared to that in *Lacticaseibacillus paracasei* Lpc-37^®^-treated animals and control rats receiving tap water. The effects of the synbiotic were evaluated using the Wallace score, the quantification of colon myeloperoxidase (MPO) and the master genes driving analgesia and inflammation. CG 1.5 alone and NCFM 10^9^ colony forming units (CFU) alone similarly decreased the visceral pain sensitivity. Lpc-37 had no significant effect. The best profile of pain perception inhibition was obtained with the combination of CG 1.5 g and NCFM 10^9^ CFU, confirming a synbiotic property. This synbiotic treatment significantly reduced macroscopic colonic lesions and MPO concentrations, and induced master genes involved in analgesia (CB1, CB2, MOR, PPARα), with a downregulation of inflammatory cytokines (IL-1β, TNFα) and an induction of IL-10 and PPARγ. In conclusion, CG 1.5 g + NCFM 10^9^ CFU significantly decreased visceral pain perception and intestinal inflammation through the regulation of master genes.

## 1. Introduction

Irritable bowel syndrome (IBS) is a common functional gastrointestinal disorder with a global prevalence in the community of 5–10%, associated with estimated annual direct and indirect costs of more than USD 20 billion/year in the USA, and is one of the leading causes of work absenteeism [1,2,3]. Although IBS represents a major burden for patients, the therapeutic strategy recommended by several gastroenterology societies (European, American, Canadian, Japanese, British societies) [4,5,6,7,8,9,10] is often inadequate, leading to the dissatisfaction of many patients with standard medical care [11,12]. The pathophysiology of IBS is complex and remains incompletely understood, involving at least in part visceral hypersensitivity, low-grade inflammation and intestinal microbiota alterations that may be modulated by prebiotics, probiotics or synbiotics [13].

Chitin–glucan is a novel dietary prebiotic that is considered a safe food ingredient by the European Food Safety Authority (EFSA) [14]. It is the major component of the cell walls of the mycelium of *Aspergillus niger* fungi, composed at 95% of a branched β-1, 3/1, 6 glucan that is linked to chitin via a β-1, 4 linkage, and is considered to be an insoluble fiber not digested or absorbed in the small intestine and that travels intact through the gastrointestinal tract to the colon [14]. EFSA recommendations concerning the intended intake of chitin–glucan in humans is 2 to 5 g/day, split into two or three doses, taken preferably with food or liquid to help it swell [14]. Previous preclinical studies in rodent models [15,16], a functional in vitro evaluation using the Simulator of the Human Intestinal Microbial Ecosystem model [17], and a clinical exploration in healthy volunteers [18] showed that the oral administration of chitin–glucan at the EFSA-recommended dosage induces the microbial signature of a prebiotic [14]. These studies found that chitin–glucan is slowly fermented in all colon segments without increasing gas production or the fecal calprotectin concentration [18,19]. Gut microbiota analysis using Illumina sequencing also revealed an increased relative abundance of the butyrate-producing genera *Roseburia* spp. and *Faecalibacterium prausnitzii*, a genus with strong anti-inflammatory properties [18,19]. Recently, additional preclinical molecular, cellular, and animal studies showed that chitin–glucan is a new generation of prebiotic able to rapidly and significantly decrease visceral perception and intestinal inflammation through the regulation of master genes for pain, inflammation, and intestinal barrier function [19]. The comparison of the visceral anti-nociceptive effect of different dosages of chitin–glucan in an animal model mimicking IBS revealed the time- and dose-dependent analgesic effect of chitin–glucan with the more rapid (3 weeks vs. 5 weeks) and more intense (50% vs. 30%) inhibition of pain perception for chitin–glucan given at a human equivalent dosage of 3 g/d versus 1.5 g/d [19]. We also demonstrated, in silico by the molecular modeling of chitin–glucan, the antimicrobial activities of the molecule chelating the most active components of gram-negative (lipopolysaccharide LPS) and -positive (lipoteichoic acid LTA) bacteria and the phospholipomannan of yeasts [19].

Probiotics, defined in 2001 by the Food and Agriculture Organization (FAO) and the World Health Organization (WHO), are live microorganisms that, when administered in “adequate amounts”, confer a health benefit on the host (FAO/WHO 2021). Concerning the use of single-strain probiotics in the treatment of IBS patients, the definitive optimal “adequate amounts” of probiotics administered orally have not been established. However, even if there is heterogeneity across studies in terms of strains, dosages and durations, most clinical trials performed in patients with IBS found that probiotics offered benefits at dosages ranging between 10^8^ and 10^10^ colony forming units (CFU)/day [20,21]. *Lactobacillus acidophilus* NCFM is a probiotic strain with a long history of use [22]. We previously showed, in a model of colorectal distension in rats, that the optimal dosage of the strain needed to decrease visceral perception was 10^9^ CFU/day when animals were treated over 15 consecutive days with dosages that increased from 10^7^ to 10^9^ CFU per day [23]. Importantly, the strain has been shown, in vitro and in animals, to reduce visceral pain perception through the induction of cannabinoid receptor 1–2 (CB1, CB2) and µ-opioid receptor (MOR) mRNA, and the induction of protein mainly expressed by intestinal epithelial cells [23]. These observations have subsequently been confirmed in human clinical research with the increased rectal expression of MOR both at mRNA and protein levels [24] and improved visceral pain perception in patients [25,26]. Also, *Lacticaseibacillus paracasei* Lpc-37 has a long history of use as a probiotic [27]. The strain has been shown to reduce anxiety in a mouse model [28] and a subsequent human clinical study [29]. The strain did, however, not induce the expression of mRNA MOR or CB receptors in vitro in HT-29 epithelial cells nor in vivo in male Sprague-Dawley rats [23].

Preclinical studies that enhance the management of IBS and increase the development of new specific targeted treatments are important. Rodent models are most commonly used to assess the pathophysiology of IBS and the development of new treatment approaches for patients with visceral pain. Among the numerous different animal models of visceral pain, TNBS-induced long-lasting visceral hypersensitivity is the reference model used for screening novel treatments for visceral pain originating within the gastrointestinal tract, demonstrating the potential to predict therapeutic efficacy [30]. In this model, rats develop mild and transitory colitis, followed by a mucosal healing period with a partial recovery of macroscopic inflammatory lesions at the end of the experiment and the persistence of hypersensitivity to colorectal distension (CRD) [30].

To evaluate whether the association of chitin–glucan and *Lactobacillus acidophilus* NCFM might be an effective synbiotic treatment option in IBS, we investigated the functional and molecular gastrointestinal responses of the compound to visceral analgesia and intestinal inflammation in rats with long-lasting visceral hypersensitivity. According to the known properties of chitin–glucan and the *Lactobacillus acidophilus* NCFM strain in the gut, we hypothesized that the simultaneous oral administration of the two compounds would have synbiotic properties in an animal model of visceral pain. We therefore compared the functional analgesic effects of chitin–glucan and/or *Lactobacillus acidophilus* NCFM alone or in combination in rats with long-lasting visceral hypersensitivity, and investigated if this association locally induced a new molecular signature of genes known to regulate the nociceptive and inflammatory responses in the gut.

## 2. Results

### 2.1. Antinociceptive Effects of the Compounds

In control rats receiving tap water, a mean pressure of 46 ± 0.9 mm Hg was required to induce pain. The treatment given throughout the study with *L. paracasei* Lpc-37 at 10^9^ CFU did not modify the pain threshold compared to that in untreated control rats except at week 7, with a modest and late increase in the pain threshold (Figure 1). In the absence of significant effectiveness with *L. paracasei* Lpc-37, no other therapeutic combination with the strain and chitin–glucan was subsequently tested.

Two weeks of treatment with chitin–glucan at 3 g/d decreased the normal visceral sensitivity, with a significant 15% increase in the pain threshold compared to untreated rats (53 ± 0.7 mm Hg vs. 46 ± 0.9 mm Hg, *p* < 0.01); meanwhile, chitin–glucan at 1.5 g/d did not cause any difference (47.5 ± 0.7 mm Hg vs. 46 ± 0.9 mm Hg, NS) (Figure 1). A similar analgesic effect of chitin–glucan at 3 g/d was observed with *L. acidophilus* NCFM at 10^9^ CFU, decreasing pain perception by 11% compared to control rats (51 ± 0.9 mm Hg vs. 46 ± 0.9 mm Hg, *p* < 0.05) (Figure 1).

Compared to control rats, animals exposed to TNBS developed a significant and long-lasting hypersensitivity that was observed 3 weeks after TNBS administration (33.9 ± 1.0 mm Hg vs. 48.0 ± 1.1 mm Hg, −29%; (*p* < 0.001) and was maintained at week 5 (33.3 ± 1.0 mm Hg vs. 48.0 ± 1.1 mm Hg, −31%; *p* < 0.001) and week 7 (33.9 ± 1.2 mm Hg vs. 50.0 ± 0.0 mm Hg, −32%; *p* < 0.001) (Figure 1 and Figure 2).

We next evaluated, in animals exposed to TNBS, the concentrations of the synbiotic combining chitin–glucan (1.5 g/d or 3 g/d) and *L. acidophilus* NCFM (10^8^ to 10^10^ CFU/d) that led to the highest and longest analgesic effect (Figure 2A,B). *L. acidophilus* NCFM at 10^8^ CFU combined with chitin–glucan did not provide an additional analgesic effect compared to chitin–glucan alone when given at 1.5 g/d (Figure 2A) or 3 g/d (Figure 2B). Similar profiles of pain perception inhibition were obtained with the two dosages of *L. acidophilus* NCFM at 10^9^ CFU or 10^10^ CFU associated with chitin–glucan at 1.5 g/g or 3 g/d (Figure 2A,B). The best profile of pain perception inhibition was obtained with the association of chitin–glucan at 1.5 g/d and *L. acidophilus* NCFM at 10^9^ CFU, confirming that the synbiotic properties of the components lead to a rapid 17% analgesic effect at week 2 (53.9 ± 1.0 mm Hg vs. 46.0 ± 0.9 mm Hg; *p* < 0.001), an increase at week 3 after TNBS administration (48.5 ± 1.2 mm Hg vs. 33.9 ± 1.0 mm Hg, 43%; *p* < 0.0001) and the maintenance of this effect at week 5 (51.0 ± 0.9 mm Hg vs. 33.3 ± 1.0 mm Hg, 53%; *p* < 0.0001) and week 7 (53.5 ± 1.5 mm Hg vs. 33.9 ± 1.2 mm Hg, 58%; *p* < 0.0001) (Figure 2A).

### 2.2. Anti-Inflammatory Effects of the Synbiotic

The anti-inflammatory effect of chitin–glucan at 1.5 g/d and *L. acidophilus* NCFM at 10^9^ CFU/d was evaluated at the end of the experiment (week 7) five weeks after TNBS administration and compared to the control animals with or without colitis receiving tap water. Compared to the control animals, the intrarectal administration of TNBS (15 mg/kg) induced significant macroscopic lesions of the colon characterized by at least two sites of inflammation and ulceration of less than 1 cm (3.7 ± 0.5 vs. 0.4 ± 0.2, *p* < 0.05; Figure 3A) and a significant increase in the level of MPO (405.7 ± 63.9 vs. 185.3 ± 7.1, *p* < 0.05; Figure 3B). Treatment with the synbiotic chitin–glucan at 1.5 g/d and *L. acidophilus* NCFM at 10^9^ CFU significantly reduced the intensity of the macroscopic colonic lesions by 35% (2.4 ± 0.4 vs. 3.7 ± 0.5, *p* < 0.05), characterized by less than five aphtoid lesions without hyperemia and colon wall thickening, and by a 49% reduction in the colonic MPO concentrations (205 ± 29 vs. 406 ± 64, *p* < 0.01) compared to rats with colitis receiving tap water (Figure 3A,B).

### 2.3. Genes mRNA Quantification in the Colon of Rats

The quantification of genes in the colon was performed at the end of the experiment (week 7) in untreated control animals without colitis (*n* = 5) and in rats with colitis treated with the synbiotic chitin–glucan at 1.5 g/d and *L. acidophilus* NCFM at 10^9^ CFU (*n* = 12) or receiving tap water (*n* = 12).

The levels of CB1, CB2 and MOR mRNA were decreased in untreated rats with colitis compared to the control animals receiving tap water, and were restored in animals with colitis receiving the synbiotic (Figure 4A–C). In rats with colitis, despite the rats receiving the symbiotic having higher rates of CB2, MOR and PPARα mRNA than the untreated animals (Figure 4B–D), only the CB1 mRNA levels increased significantly (78.6 ± 12.7 vs. 19.7 ± 4.5, +398%; *p* < 0.05) (Figure 4A).

A significant increase in IL-1β (17.5 ± 7.6 vs. 2.3 ± 0.6, *p* < 0.05) and TNFα (4.6 ± 2.0 vs. 1.5 ± 0.5, *p* < 0.05) mRNA was found in untreated rats with colitis compared to the control animals with restored levels of IL-1β (17.0 ± 2.7 vs. 17.5 ± 7.6, *p* < 0.005) and in the TNFα mRNA levels (1.4 ± 0.4 vs. 4.6 ± 2.0, *p* < 0.005) of rats with colitis receiving the synbiotic (Figure 5A,B). Concerning genes with anti-inflammatory effects, significantly increased levels of IL-10 (3.8 ± 2.0 vs. 0.9 ± 0.3, *p* < 0.05) and PPARγ (0.8 ± 0.0 vs. 0.6 ± 0.0, *p* < 0.05) mRNA were observed in the colon of rats with colitis treated with the synbiotic compared to untreated animals (Figure 5C,D).

## 3. Discussion

Studies on the basic molecular mechanisms that enhance IBS management and facilitate the development of new specific targeted treatments are important. Visceral hypersensitivity is a key feature of IBS. In the present study, we investigated the potential synbiotic properties of a combination of chitin–glucan and the probiotics *L. acidophilus* NCFM or *L. paracasei* Lpc-37 in a model of long-lasting colon hypersensitivity induced by TNBS. We show the marked synergistic analgesic and anti-inflammatory effects of chitin–glucan and *L. acidophilus* NCFM, suggesting that this synbiotic is worth exploring in patients with IBS.

To determine visceral hypersensitivity, we used CRD, the method most widely used for assessing visceral sensitivity in rodents due to its good sensitivity and robust reproducibility [23,31]. We showed that chitin–glucan, a novel dietary prebiotic used in humans at a recommended dosage of 1.5 g/d, in association with the probiotic strain of *L. acidophilus* NCFM, known to reduce visceral perception in preclinical studies and human clinical research [23,24,25,26], decreased visceral perception by 58% seven weeks after TNBS administration. The synbiotic combining chitin–glucan at 1.5 g/d and *L. acidophilus* NCFM at 10^9^ CFU/d showed the best profile of pain perception inhibition, resulting in a more rapid and 228% superior analgesic effect compared to chitin–glucan alone at 1.5 g/d after nine weeks of administration. While the analgesic effect of chitin–glucan alone was time- and dose-dependent, with greater efficacy at 3 g/d than 1.5 g/d, combination with *L. acidophilus* NCFM did not show any difference for these two dosages of chitin–glucan. Concerning the use of probiotics in IBS, it is known that their benefits on abdominal pain are strain- and dose-dependent [32,33]. In the present study, the association of *Lacticaseibacillus paracasei* Lpc-37 with chitin–glucan had no significant effect on visceral pain perception, and the best antinociceptive property of the synbiotic was obtained with *L. acidophilus* NCFM at 10^9^ and 10^10^ CFU/d, corresponding to the dosages usually effective for probiotics in most clinical trials [34].

Both intestinal hypersensitivity and low-grade mucosal inflammation are relevant targets for the control of IBS symptoms [35,36]. Low-grade mucosal inflammation is a potentially important contributor to IBS pathogenesis, but so far there is a lack of efficacious treatment approaches targeting this mechanism [37]. In rats with long-lasting visceral hypersensitivity induced by the intra-rectal administration of TNBS, the active inflammation of the colon remains present for a few days after TNBS exposure, followed by a partial recovery period of 8 weeks where low-grade inflammation and hypersensitivity to CRD persist [30]. To gain insight into the mechanisms underlying the therapeutic efficacy of our synbiotic, we showed that the combination of chitin–glucan at 1.5 g/d and *L. acidophilus* NCFM at 10^9^ CFU/d decreased colonic inflammation, reducing the intensity of macroscopic lesions by 35% and the levels of colonic MPO, considered as a marker of neutrophil infiltration, by 49% compared to untreated rats with colitis.

The development of synbiotics is motivated by the potential ability of the prebiotic to enhance the growth and metabolic activities of the probiotic [38]. At this stage of the development of our synbiotic, we do not know if chitin–glucan is preferentially metabolized by *L. acidophilus* NCFM and can support its growth. *L. acidophilus* NCFM has been reported to survive the rat gastrointestinal tract [23] and a symbiotic interaction is therefore possible. We know that both chitin–glucan and *L. acidophilus* NCFM have individual analgesic and anti-inflammatory properties involving at least in part, respectively, cannabinoid and µ-opioid receptors and a restoration of the disbalance between IL-10 and IL-1β, IL-8 and TNFα [19,23,39]. In the present study, in addition to genes known to be involved independently by chitin–glucan and *L. acidophilus* NCFM in the regulation of pain and inflammation, we found significantly increased levels of PPARα and PPARγ mRNA in the colon of rats receiving the synbiotic compared to untreated animals. PPARα and γ are two nuclear receptors highly expressed in the colon, particularly by epithelial cells [40,41]. They are considered to play a key role in gut homeostasis, particularly in the regulation of visceral hypersensitivity and intestinal inflammation [40,41,42,43,44]. Even if the significantly increased levels of PPARα and γ mRNA are not always associated with increased receptor activation, these data indicate that the synbiotic chitin–glucan and *L. acidophilus* NCFM induced a new molecular signature involving the G protein-coupled receptors PPARα and PPARγ, which may be involved in the modulation of inflammation-induced persistent visceral pain signaling.

Previous studies in hamsters fed with an atherogenic diet [15] and mice with metabolic alterations induced by a high-fat diet [16] support the beneficial effects of chitin–glucan on the composition of the gut microbiota and cardiometabolic profile with respect to the development of obesity, associated metabolic diabetes and hepatic steatosis. Similarly, an improvement in the postprandial glycemic and lipemic profiles was observed after a 3-week supplementation of chitin–glucan at 4.5 g/day in subjects at cardiometabolic risk (BMI: 25–35 kg/m^2^) [45]. Obesity and particularly visceral obesity have an increasing prevalence in patients with inflammatory bowel diseases [46,47]. Visceral obesity is associated with an increased risk of developing Crohn’s disease [48] and may have a negative impact on the natural course of the disease, complications and the outcomes of medical and surgical therapies [47,49]. In the present study, we included 5-week-old rats with normal weight (175–200 g). A similar pattern of weight evolution was observed in rats with colitis receiving tap water (control group), chitin–glucan, *L. acidophilus* NCFM or *L. paracasei* Lpc-37. Apart from the increased levels of canonical activators of lipid uptake and adipogenesis PPAR α and γ mRNA observed in the colon of rats receiving chitin–glucan and *L. acidophilus* NCFM compared to the control animals with colitis, our study did not evaluate the influence of chitin–glucan supplementation on the composition of the gut microbiota and cardiometabolic profiles of animals with long-lasting colon hypersensitivity. Further studies in animals fed with normal and high-fat diets will be needed to explore this possibility.

## 4. Conclusions

The simultaneous oral administration of chitin–glucan and *Lactobacillus acidophilus* NCFM resulted in a twofold superior visceral analgesic effect compared to the administration of the prebiotic or probiotic alone, suggesting the synbiotic properties of our compounds. This synbiotic combining chitin–glucan at 1.5 g/d and *L. acidophilus* NCFM at 10^9^ CFU provided also a direct therapeutic benefit, decreasing colonic inflammatory lesions, locally modulating the expression of genes involved in the regulation of pain and inflammation and encoding for opioid and cannabinoid receptors, inflammatory and anti-inflammatory cytokines and the nuclear receptors PPAR α and γ. These advances highlight the ability of our synbiotic to target two important pathophysiological mechanisms of IBS and its therapeutic potential as a promising next-generation natural non-pharmacological treatment for patients with IBS or IBS-like symptoms. A double-blind, randomized, clinical trial in patients with IBS receiving the synbiotic chitin–glucan at 1.5 g/d and *L. acidophilus* NCFM at 10^9^ CFU is now in progress (Protocol BK-IBS-2301. EudraCT Nr: B7072023000058. File 2023-179).

## 5. Materials and Methods

### 5.1. Chitin–Glucan and Bacteria

Chitin–glucan from the cell walls of *Aspergillus niger* was provided by Kitozyme SA (Herstal, Belgium). *Lactobacillus acidophilus* NCFM^®^ and *Lacticaseibacillus paracasei* Lpc-37^®^ were provided by IFF Health (Madison, WI, USA). Rodents received chitin–glucan in combination or not with *L. acidophilus* NCFM or *L. paracasei* Lpc-37 by oral gavage once per day for 9 weeks. Chitin–glucan was used at a dose of either 25 mg/kg body weight (BW)/d or 50 mg/kg BW/d, corresponding to a human equivalent dose (HED) of, respectively, 1.5 g/d and 3.0 g/d for a 70 kg man [15]. *L. acidophilus* NCFM and *L. paracasei* Lpc-37 were used at doses increasing from 10^8^ to 10^10^ colony-forming units (CFU) per day.

### 5.2. Model of TNBS-Induced Long Lasting Visceral Hypersensitivity in Rats

#### 5.2.1. Rats

Animal experiments were performed in accredited facilities at Institut Pasteur in Lille according to governmental guidelines. All the studies were approved by the local investigational ethics review board (Nord-Pas-de-Calais CEEA N°75, Lille, France; protocol reference numbers 352,012 and 19-2009R) and French government agreement n° APAFIS#16100-2018070309443695 v4 (colorectal distension) and APAFIS#9148-201901101416384 v1 (colitis). Animals were housed three per cage and had free access to standard rodent chow (Safe A04 P2,5; SAFE, Augy (France)) and tap water.

Male Sprague Dawley rats, 5 weeks old and weighing 175 to 200 g, were obtained from Janvier labs (Saint-Berthevin, France). The rats were randomized into different groups by a manual procedure and acclimatized to the study conditions for a period of at least 7 days before the beginning of the pre-treatment period. Upon completion of the treatment at week 7, the animals were euthanized by cervical dislocation after gaseous anesthesia (Isoflurane).

#### 5.2.2. TNBS-Induced Visceral Hypersensitivity

The rats were anesthetized for 2 h using a subcutaneous injection of xylazine at 12.5 mg/kg (Bayer, Rompun 2%)/ketamin at 25 mg/kg, (Virbac, Ketamin 1000 (100 mg/mL). Colitis was induced by a colonic injection of 250 µL of TNBS (15 mg of TNBS corresponding to 80 mg/kg of body weight dissolved in a 1:0.4 mixture of 0.9% NaCl with 100% EtOH) located 7–8 cms upstream of the anus using a catheter (Centracath, ref. 1230.20, 8 cm of length). Control rats received 250 µL of the solution without TNBS (1:0.4 mixture of 0.9% NaCl with 100% EtOH) using the same technique.

#### 5.2.3. Evaluation of Visceral Pain by Colorectal Distension

Nociception in animals was assessed by measuring the intracolonic pressure required to induce a behavioral response during CRD due to the inflation of a balloon introduced in the colon. This response is characterized by an elevation of the hind part of the animal’s body and clearly visible abdominal contraction corresponding to severe contractions [23,50,51]. Briefly, the rats were anesthetized with volatile anesthesia (2% isoflurane), the balloon [23,50,51] was inserted intra-rectally in a minimally invasive manner at 7 cm from the anus, and the catheter was taped to the base of the tail. After 5 min, the rats were placed in the middle of a 40 × 40 cm Plexiglas box and the catheter was connected to an electronic barostat apparatus (Distender Series IIR™, G&J Electronics), Toronto, ONT, Canada. An increasing pressure was continuously applied until a pain behavior was displayed or a cut-off pressure of 80 mm Hg was reached.

### 5.3. Experimental Design

Animals (*n* = 129) were weighted and randomly distributed into 12 groups: one control group (*n* = 5) and eleven groups with colitis, including a negative control group receiving tap water (*n* = 12), two groups of rats treated daily by oral gavage of chitin–glucan alone at 1.5 g/d (*n* = 12) or 3 g/d (*n* = 10), two groups receiving *L. acidophilus* NCFM at 10^9^ CFU/d (*n* = 12) or *L. paracasei* Lpc-37 (*n* = 12) at 10^9^ CFU/d), three groups of 12 rats receiving chitin–glucan at 1.5 g/d in association with *L. acidophilus* NCFM (10^8−9−10^ CFU/d) and three groups of 10 rats receiving chitin–glucan at 3 g/d in association with *L. acidophilus* NCFM (10^8−9−10^ CFU/d) (Figure 6).

Visceral sensitivity was assessed at regular intervals throughout the 9 weeks of the experiment, with a first evaluation at week -2 corresponding to the basal condition, an evaluation at week 0 just before colitis induction, and at weeks 3-5-7 after colitis induction (Figure 1, Figure 2 and Figure 3).

### 5.4. Evaluation of Macroscopic Inflammatory Lesions (Wallace Score)

At the end of the experiment, macroscopic visible damage of the opened colon was evaluated independently and blindly by two operators under a dissecting microscope at a magnification of 5. Lesions were scored on a 0–10 scale using the Wallace scoring method based on criteria reflecting different types of inflammation, such as hyperemia, ulcers, the thickening of the bowel and the extent of ulceration [41] (Table 1).

### 5.5. Quantification of Colonic Myeloperoxidase (MPO)

For the myeloperoxidase (MPO) assay, each colon sample located precisely 2 cm above the anal canal was homogenized in a buffer containing 200 mM of NaCl, 10 mM of Tris Base, 5 mM of EDTA and 1 mM of PMSF with an Ultra Turax T10 (IKA, Staufen, Germany). The homogenates were then centrifuged at 1500 g for 15 min at 4 °C to pellet the insoluble debris. The total protein concentration was measured in the supernatants with the Bradford method (Bradford MM. Anal Biochem 1976) using the Quick Start Bradford Protein Assay (Bio-Rad, Hercules, CA, USA). The MPO levels in the colonic specimens were then quantified using the rat MPO ELISA kit from Hycult Biotech (Uden, The Netherlands) according to the manufacturer’s instructions. The results are expressed as MPO (ng)/total protein (mg) ratios.

### 5.6. mRNA Quantification in Colon Samples

The colonic specimens were frozen at −80 °C; then, the total RNA was extracted using the Nucleospin RNA kit (Macherey Nagel). The main analgesic-related genes (MOR, CB1, CB2, PPARα) and inflammatory-related genes (IL-1β, TNFα, IL-10, PPARγ) were assessed by quantitative RT-PCR (Table 2). Briefly, the total RNA was extracted using the Nucleospin RNA Kit (Macherey–Nagel, Hoerdt, France). After RNAse inactivation, the total RNA was cleaned of traces of genomic DNA via DNase treatment and eluted in RNAse- and DEPC-free water. The RNA purity was evaluated by UV spectroscopy using a Nanodrop system at wavelengths ranging from 220 to 350 nm. One microgram of total RNA was used to perform quantitative RT-PCR using LightCycler FastStart DNA Master SYBR Green I (Roche Diagnostics, Indianapolis, IN, USA), according to the manufacturer’s protocol. The sequences and relative NCBI reference sequences of the primer sets are listed in Table 2. For each reaction, a critical threshold cycle (Ct) value indicating the cycle number at which the DNA amplification should be performed was determined. The relative gene expression value was calculated as E = 2^−ΔCt^, where ΔCt is the difference in the crossing points between the housekeeping gene glyceraldehyde-3-phosphate dehydrogenase (GAPDH) and each gene.

### 5.7. Statistical Methods

Quantitative variables are described as means ± standard error of the means. Categorical variables are expressed as percentage and frequency. Intragroup analyses were conducted using the two-tailed paired t-test or Wilcoxon signed-rank test (non-parametric test comparing ranks) depending on the distribution of the variable of interest for continuous variables, comparing baseline values with the values recorded at W0-3-5-7. Comparisons between groups were performed using the Student t-test or Mann–Whitney–Wilcoxon test (non-parametric test comparing ranks) depending on the distribution of the variable of interest. Statistics were calculated using StatXact9 CrossOver software (Cytel Inc., Cambridge, MA, USA). All statistical tests were two-sided and differences were considered statistically significant if the *p* value was <0.05.

## Figures and Tables

**Figure 1 ijms-25-10732-f001:**
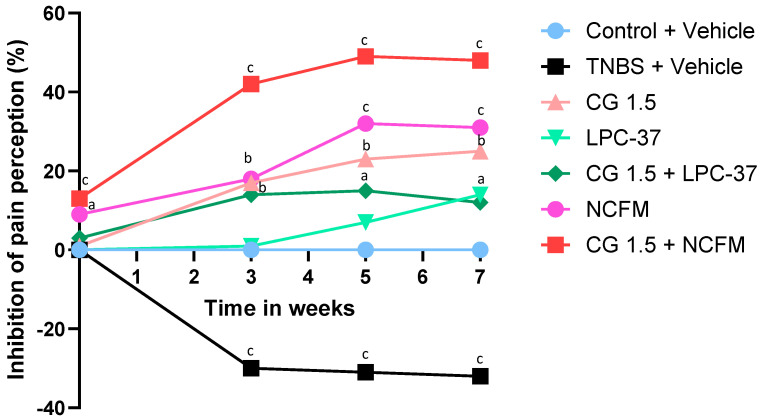
Time-related analgesic effects of chitin–glucan, *L. acidophilus* NCFM and Lpc-37^®^. Inhibition of pain perception in % at week (W) 0-3-5-7 compared to W-2 in untreated animals receiving tap water (blue), untreated animals sensitized by TNBS (black), TNBS-sensitized rats treated with chitin–glucan at 1.5 g/d (light pink), TNBS-sensitized rats treated with *L. acidophilus* NCFM at 10^9^ CFU/d (dark pink), Lpc-37^®^ (light green), TNBS-sensitized rats treated with chitin–glucan at 1.5 g/d and *L. acidophilus* NCFM at 10^9^ CFU/d (red), and TNBS-sensitized rats treated with chitin–glucan at 1.5 g/d and Lpc-37^®^ (dark green). a: *p* < 0.05; b: *p* < 0.01; c: *p* < 0.001. CG: chitin–glucan.

**Figure 2 ijms-25-10732-f002:**
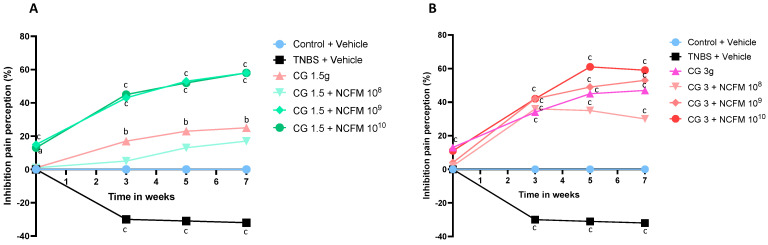
Time- and dose-related analgesic effects of the association of chitin–glucan and *L. acidophilus* NCFM. (**A**) Inhibition of pain perception in % at week (W) 0-3-5-7 compared to W-2 in untreated animals receiving tap water (blue), untreated animals sensitized by TNBS (black), TNBS-sensitized rats treated with chitin–glucan at 1.5 g/d (pink), TNBS-sensitized rats treated with chitin–glucan at 1.5 g/d and *L. acidophilus* NCFM at 10^8^ CFU/d (light green), TNBS-sensitized rats treated with chitin–glucan at 1.5 g/d and *L. acidophilus* NCFM at 10^9^ CFU/d (medium green), and TNBS-sensitized rats treated with chitin–glucan at 1.5 g/d and *L. acidophilus* NCFM at 10^10^ CFU/d (dark green). (**B**) Inhibition of pain perception in % at week (W) 0-3-5-7 compared to W-2 in untreated animals receiving tap water (blue), untreated animals sensitized by TNBS (black), TNBS-sensitized rats treated with chitin–glucan at 3 g/d (purple), TNBS-sensitized rats treated with chitin–glucan at 3 g/d and *L. acidophilus* NCFM at 10^8^ CFU/d (light pink), TNBS-sensitized rats treated with chitin–glucan at 3 g/d and *L. acidophilus* NCFM at 10^9^ CFU/d (medium pink), and TNBS-sensitized rats treated with chitin-glucan at 3 g/d and *L. acidophilus* NCFM at 10^10^ CFU/d (red). a: *p* < 0.05; b: *p* < 0.01; c: *p* < 0.001. CG: chitin–glucan.

**Figure 3 ijms-25-10732-f003:**
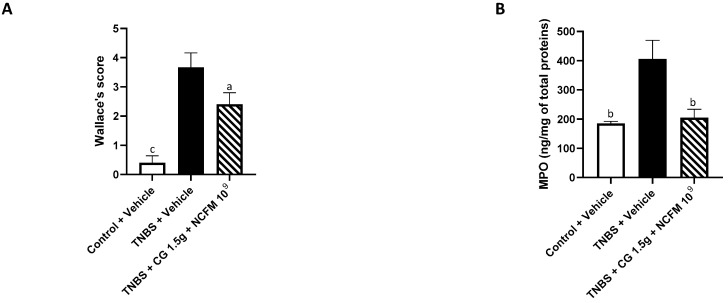
Improvement in Wallace score for macroscopic inflammatory lesions and colonic levels of MPO in animals receiving the symbiotic. (**A**) Wallace score in untreated control rats (control + vehicle, white) and in rats with colitis receiving tap water (TNBS + vehicle, black) or the symbiotic (TNBS + CG 1.5 g + NCFM 10^9^, hatched bars). (**B**) Myeloperoxidase (MPO) levels in the colon of untreated control rats (control + vehicle, white) and in rats with colitis receiving tap water (TNBS + vehicle, black) or the symbiotic (TNBS + CG 1.5 g + NCFM 10^9^, hatched bars). a: *p* < 0.05; b: *p* < 0.01; c: *p* < 0.001. CG: chitin–glucan.

**Figure 4 ijms-25-10732-f004:**
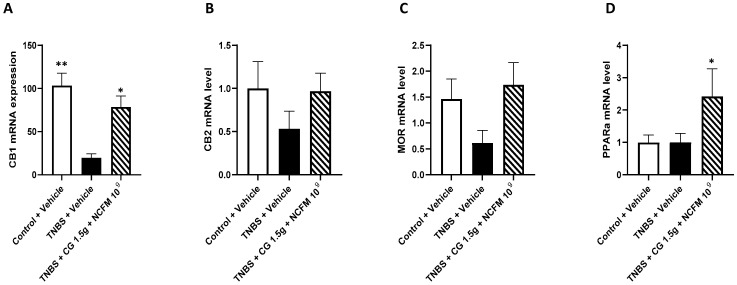
mRNA levels of analgesic-related receptors in the colon of rats treated with the symbiotic. Levels of CB1 (**A**), CB2 (**B**), MOR (**C**) and PPARα (**D**) mRNA in the colon of untreated control rats (control + vehicle, white), rats with colitis receiving tap water (TNBS + vehicle, black) and rats with colitis treated with the symbiotic (TNBS + CG 1.5 g + NCFM 10^9^, hatched bars). *: *p* < 0.05, **: *p* < 0.01. CG: chitin–glucan.

**Figure 5 ijms-25-10732-f005:**
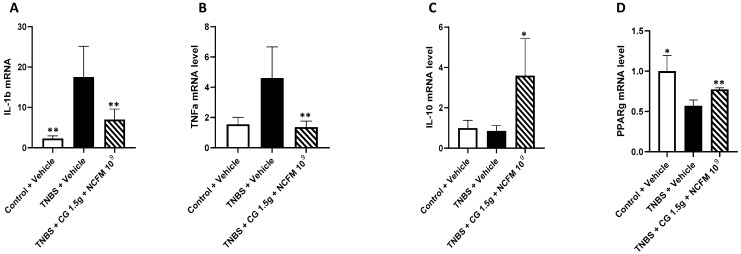
mRNA levels of inflammatory-related genes in the colon of rats treated with the symbiotic. Levels of IL-β (**A**), TNFα (**B**), IL-10 (**C**) and PPARγ (**D**) mRNA in the colon of untreated control rats (control + vehicle, white), rats with colitis receiving tap water (TNBS + vehicle, black) and rats with colitis treated with the symbiotic (TNBS + CG 1.5 g + NCFM 10^9^, hatched bars). *: *p* < 0.05; **: *p* < 0.01. CG: chitin–glucan.

**Figure 6 ijms-25-10732-f006:**
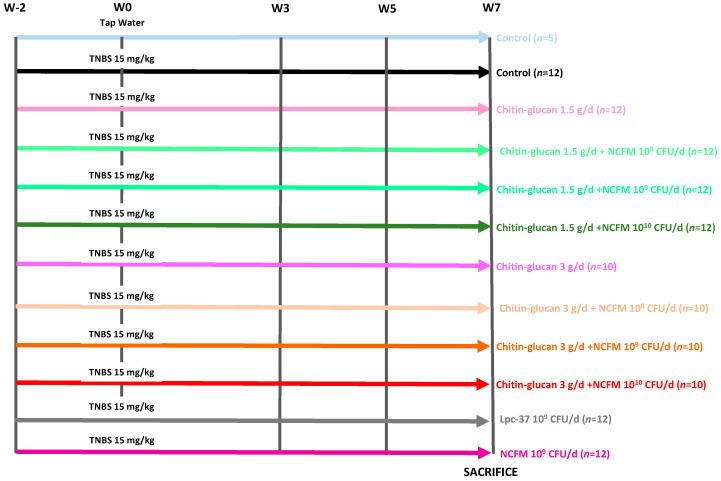
Long-lasting visceral hypersensitivity in rats: study design. Analgesic effects of the compounds evaluated by pain threshold at week (W) -2-0-3-5-7.

**Table 1 ijms-25-10732-t001:** Wallace score (0–10).

Score	Criteria of Macroscopic Evaluation
0	No inflammation
1	Hyperemia without ulcerations
2	Hyperemia with thickening of the mucosa without ulcerations
3	1 ulceration without thickening of the colonic wall
4	2 or more ulcerative or inflammatory sites
5	2 or more ulcerative or inflammatory sites with an extent > 1 cm
6	Ulcerative or inflammatory site > 2 cm
7	Ulcerative or inflammatory site > 3 cm
8	Ulcerative or inflammatory site > 4 cm
9	Ulcerative or inflammatory site > 5 cm

**Table 2 ijms-25-10732-t002:** Sequences of the primers.

Rat Genes	Primer Sequences (5′ → 3′)
GAPDH	F:5′-CTG-TTC-TAG-AGA-CAG-CCG-CAT-CT-3′ R: 5′-ACA-CCG-ACC-TTC-ACC-ATC-TTG-3′
IL-1b	S: 5′-TgAAAgCTCTCCACCTCAATggAC-3′ AS: 5′-TgCAgCCATCTTTAggAAgACACg-3′
IL-10	S: 5′-CAgTCAgCCAgACCCACAT-3′ AS: 5′-gCTCCACTgCCTTgCTTT-3′
TNF-α	S: 5′-AgCACAgAAAgCATgATCCgAg-3′ AS: 5′-CCTggTATgAAgTggCAAATCg-3′
PPARγ	F: 5′-CTgACCCAATggTTgCTgATTAC-3′ R: 5′-ggACgCAggCTCTACTTTgATC -3′
PPARα	F: 5′-ACgATgCTgTCCTCCTTgATg-3′ R: 5′-gTgTgATAAAgCCATTgCCgT-3′
MOR	F: 5′-TTCTgCATTgCTTTgggTTACACg-3′ R: 5′-CTgACAgCAACCTgATTCCACgTA-3′
CB1	F: 5′-ATgAAgTCgATCCTAgATggCCTTg-3′ R: 5′-gCTCCCCACACTggATg-3’
CB2	F: 5′-gATAgCTCggATgCggCTAgACgTg-3′ R: 5-CAgCATggggCTgTggATCgAgg-3′

## Data Availability

The original contributions presented in the study are included in the article, further inquiries can be directed to the corresponding author.

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
