# Peer review of "A Synbiotic Combining Chitin–Glucan and Lactobacillus acidophilus NCFM Induces a Colonic Molecular Signature Soothing Intestinal Pain and Inflammation in an Animal Model of IBS"

_ijms, 2024, doi:10.3390/ijms251910732_

Round 1

Reviewer 1 Report

Comments and Suggestions for Authors

The objective of the study conducted by Capirchio and colleagues was to assess the impact of a combination of two prebiotics, namely chitin-glucan and Lactobacillus acidophilus NCFM, on the gastrointestinal response in an induced colitis model. The secondary objective of this study is to assess the potential clinical applications of the two products in a commercial setting. The selection of experimental models and the scope of the analyses performed are appropriate and do not give rise to any objections. However, the work contains a rather chaotic layout and does not adhere to the recommendations of the journal, which makes it challenging to read.

Specific comments:

Affiliation: Please adhere to the guidelines set forth by the journal. Please add superscript next to the names.

In the abstract, please remove the phrase "200 words."

On page 2, please remove the entire page.

On page 3, the term expression refers to genes. It is imperative that the authors ensure the precise usage of this term in the intended context.

The paper on page 3 presents the goal without a hypothesis. It should be noted that the hypothesis differs from the purpose of the study.

Page 4 - The distance of 8 cm from the anus is unclear. Please clarify how this was performed. Was it done in a straight line, 8 cm from the anus?

Page 7 - The statistical description is insufficient. For example, it is unclear whether a normal distribution was tested and how.

Conclusion - The conclusion is not sufficiently detailed. It is simply a reiteration of the results of the work described.

Reviewer 2 Report

Comments and Suggestions for Authors

The research article titled " Modulation of intestinal pain and molecular signature by a synbiotic combining chitin-glucan and Lactobacillus acidophilus NCFM" authored by Capirchio et al. The authors investigated the potential of Chitin-glucan (CG)  a new generation of prebiotic. and Lactobacillus acidophilus NCFM.  The study was well-planned, the results were presented logically, and the discussion was grounded in the existing literature. The authors concluded that Chitin-glucan and NCFM significantly reduced visceral pain perception and intestinal inflammation by regulating master genes. However, these comments need to be addressed.

Comments:

1.    If chitin-glucan (1.5g/d) is administered via oral gavage, how much of it will reach the colon to be effective against TNBSE colitis?

2.    Did the authors check when administering L. acidophilus NCFM 10^9 via oral gavage, it is crucial to determine the amount that will reach the colon to effectively combat colitis.

3.    How did the authors decide on the chitin-glucan and L. acidophilus NCFM dosages? Did they perform dose-dependent studies for both? If so, provide the data in the supplementary material.

4.    Figures 5 and 6: The authors used letters to indicate significance. Can that be changed to * asterisk, and single, double, and triple to denote different significance levels? As it will be easy to understand.

5.    Did the authors measure the levels of IL, TNF-α, and PPARs in the serum of all groups?

6.    Does chitin-glucan and NCFM play a role in obesity-associated colitis? If so, how can this study be linked to obesity? Consider including this in the discussion.

Round 2

Reviewer 1 Report

Comments and Suggestions for Authors

The authors provided responses to my inquiries that were, for the most part, satisfactory.

Author Response

The reviewer 1 seems satisfied by our point-by-point responses (see below).

I don't see any additional response to add ?